# COVID-19 and screen-based sedentary behaviour: Systematic review of digital screen time and metabolic syndrome in adolescents

Sarah Musa[1]*, Rowaida Elyamani[2], Ismail Dergaa[1]

1 Department of Preventative Health, Primary Health Care Corporation, Doha, Qatar, 2 Department of Medical Education, Hamad Medical Corporation, Doha, Qatar

* smusa@phcc.gov.qa

**Data Availability Statement:** All relevant data are within the paper and its Supporting information files.

## Abstract

### Aim

The COVID-19 pandemic has prompted governments around the globe to implement various restriction policies, including lockdown, social distancing, and school closures. Subsequently, there has been a surge in sedentary behaviour particularly screen time (ST) together with a significant decline in physical activity that was more marked amongst children and adolescents. Excessive screen exposure in adolescents has been correlated with cardio-metabolic risk factors including obesity, hypertension, high cholesterol, and glucose intolerance that may have adverse morbidity and mortality implications in adulthood. Thus, the current study aimed to synthesize the literature on the relationship between ST of various types and the risk of metabolic syndrome (MetS) in adolescents in the context of the COVID-19 pandemic.

### Methods

In August 2021, a systematic search of the literature was undertaken using electronic databases: PubMed, PsycINFO, and the Cochran library. Studies were considered if they met the following key eligibility criteria: (i) Measure of ST as an exposure (TV, computer, videogames, internet, smartphone, tablet), using quantified duration/frequency either self-reported or observed; (ii) Measure of MetS as an outcome with standard definition and/or criteria required to establish MetS diagnosis. The Quality Assessment Tool for Observational Cohort and Cross-Sectional Studies was used to assess the risk of bias.

### Results

A total of ten studies met the inclusion criteria, and the majority were cross sectional studies. Most studies met fair bias scoring. Overall, the review revealed considerable evidence that suggests a significant negative association between ST and components of MetS among adolescents with dose-response association.

**Funding:** The publication of this article was funded by Qatar National Library. The funders had no role in study design, data collection and analysis, decision to publish, or preparation of the manuscript.

**Competing interests:** The authors have declared that no competing interests exist.

## Conclusion

During the pandemic, screen usage may become more prevalent through periods of school closures, lockdowns, social isolation, and online learning classes. Public health policies and health promotion strategies targeting parents are needed to raise awareness of the adverse health effects associated with screen-based sedentary behaviour as a precursor of NCDs. Parent or home focused interventions might be effective in limiting adolescents' screen exposure, alternatively substituted with an appropriate level of physical activity.

## PROSPERO registration number

PROSPERO 2021 CRD42021272436.

## Introduction

The ongoing COVID-19 pandemic has impacted billions of children's and adolescents' lives in an unprecedented manner [1–3]. To limit the spread of the virus, stringent preventative measures were imposed worldwide. Countries around the globe announced complete lockdown which included school closures for the most part of the year 2020 and extended partially into the following years. Around 1.5 billion children (aged 5–12 year) and youths (aged 13–17 year) were transitioned into virtual learning [4, 5]. Disruption of daily routines, limited mobility and social constraints have considerably increased engagement in sedentary activities especially screen time (ST) [6, 7]. Available evidence indicates that screen-based sedentary behaviours have been associated with unhealthy dietary habits [6], interrupted sleep patterns [8] and limited opportunities for children and adolescents to engage in physical activity [9], all of which comprise a combination of risk factors for metabolic syndrome.

Given the revolutionary advances in digital technologies, the question of how to adequately classify ST remains a challenge [10]. The World Health Organization (WHO) defines ST as "Time spent passively watching screen-based entertainment (TV, computers, mobile devices)," excluding other innovative and modern forms of virtual realities, interactive video-gaming where physical activity or movement is required [11].

The COVID-19 pandemic has caused a marked increase in ST across the globe. A large observational study (n = 8395) in 10 European countries revealed that 69.5% [95%CI: 68.5–70.5] of young adolescents aged 6–18 years have exceeded the recommended limit of ST (>2 h/day) during weekdays and 63.8% during weekend [95%CI: 62.7–64.8]. Children residing in mildly affected countries and those in countries with lower level of restrictions were less likely to exceed that limit (OR = 3.25 [95%CI: 2.38–4.45) and OR = 1.42 [95% CI: 1.07–1.90], respectively) [12]. Similarly, findings from (ABCD) study during the early stages of the pandemic reported a mean (SD) of 7.70 (5.74) h/day of screen use, a more than twofold increase as compared to the pre-pandemic figure [13].

MetS is defined as a set of cardio-metabolic risk factors that includes glucose intolerance, central obesity, hypertension, and dyslipidaemia [14]. Lifestyle factors such as insufficient moderate-to-vigorous physical activity (MVPA), low cardiorespiratory fitness, smoking, and sedentary behaviour are amongst the various possible predictors of MetS in adolescents [15]. According to the American Heart Association (AHA), National Heart, Lung and Blood Institute (NHLBI) and International Diabetes Federation (IDF), the diagnosis of MetS is based on the presence of three of the followings: waist circumference (WC) indicative of central obesity

(at least 102 cm in men and 89 cm in women), raised triglyceride (<40 mg/dl) in males, <50 mg/dl in females), raised blood pressure (systolic BP≥ 130 or diastolic BP ≥ 85 mmHg or receiving treatment for hypertension), and raised fasting glucose level (≥100 mg/dL, or diagnosed with type 2 diabetes) [16]. The diagnosis of MetS is usually established after the age of 10 years. In older children and adolescents aged 10–16 years, MetS is diagnosed in the presence of central adiposity (≥90th) and two of the following: triglycerides (TG)≥ 150 mg/dl, HDL-C <40 mg/dl, systolic blood pressure (SBP) ≥ 130 mmHg or diastolic blood pressure (DBP) ≥85 mmHg, fasting plasma glucose (FG) ≥ 100 mg/dl or previously diagnosed type 2 diabetes [17].

MetS in children and adolescents has become a major public health concern, with prevalence reaching as high as 38.9% in the general population and relatively higher in overweight/obese children [18]. Numerous studies [19–21] have suggested that metabolic risk factors in childhood are associated with an increased risk of type 2 diabetes, subclinical atherosclerosis, and cardiovascular disease in adulthood. The pathological process underlying MetS begins in childhood with complex interrelated genetic and environmental factors [22]. Screen-based behaviours and physical inactivity are linked to higher levels of inflammatory biomarkers like interleukin-6 (IL-6) and tumour necrosis factor- (TNF-), which stimulate C-reactive protein (CRP), an important causative pathway leading to dyslipidaemia, insulin resistance, and cardiovascular disease [23]. A study by Strizich et al. showed that lower levels of MVPA were associated with higher glucose/lipid profiles and increased inflammatory biomarkers [24].

According to the current physical activity (PA) guidelines, children and adolescents should be engaged in at least 60 minutes of MVPA and no more than two hours of sedentary recreational ST daily [25]. Nevertheless, restrictions imposed due to the COVID-19 pandemic as well as prolonged missed opportunities in physical education due to school closures are foreseeable to profoundly limit the ability to meet these recommendations.

The association between ST and MetS among adolescents has been investigated in several studies prior to the declaration of the COVID-19 pandemic [26, 27]. However, results were found inconclusive for the most part owing to limited data and generalizability of findings to different types of ST considering the duration, content, and context of exposure [28, 29]. In a recent systematic review, authors pointed out limitations in approving the direct cause and effect relationship between excessive ST and MetS in adolescents [30]. For instance, de Oliveira RG et al. [31] revealed that ST of more than 2h/day during the weekend was significantly associated with a twofold increased risk of MetS, while insignificant association was observed concerning other days of the week. Khan et al. [32], however, observed a positive linear correlation between ST and MetS in children and adolescents, indicating a dose-response relationship for every 2 hours/day increase in ST (OR: 1.29, 95 percent CI 1.12–1.46) [32]. Stiglic and Viner [33] in their review found inadequate and unreliable evidence of association between ST and MetS, though, higher levels of ST were related significantly to an increased caloric intake, lower nutritional food, being obese, and having an overall reduced quality of life.

In light of the evolving pandemic, the prolonged screen-based sedentary behaviour exacerbated by remote learning remains a particular cause of concern. The emergence of MetS in earlier life indicates a serious risk of persistence into adulthood. Identification of contributing risk factors is of a great importance to planning for cost-effective prevention strategies. Therefore, an updated evaluation of available evidence is needed to examine the association between ST and MetS (with dose-response gradient) among adolescents, taking into consideration adjustment of potential confounders such as PA and dietary behaviour.

Thus, the aim of the current systematic review was to summarise the findings of studies that have looked at the quantifiable association between various forms of ST and MetS in adolescents aged 12 to 18 years.

## Methods

### Protocol and registration

We followed the Preferred Reporting Items for Systematic Reviews and Meta-Analyses (PRISMA) statement for reporting this systematic review [34] as shown in S1 Table. The protocol was registered with PROSPERO (CRD42021272436).

### Search strategy

A systematic search strategy was conducted using the following electronic bibliographic databases to identify relevant studies: PubMed Central/MEDLINE, Cochrane Library, PsycINFO and google scholar without the use of a filter to limit the date of publication or language. The search was conducted between August 2021 and September 2021. Only currently open access published articles were retrieved. The following keywords were used for the search: "screen time" OR "sedentary behaviour" OR "television" OR "computer" OR "internet" OR "video-games" AND "MetS" OR "cardiometabolic" OR "obesity" AND "adolescents" OR "children" OR "youth" OR "school-aged". Titles and abstracts of potentially relevant articles were screened by one reviewer to assess relevance and suitability for inclusion. Full-text articles with reference lists were retrieved and examined for appropriateness. Another reviewer backtracked all reviewed articles for double-checking. Any discrepancies or disagreements between reviewers were resolved by either discussion or a third reviewer. RefWorks software was used to remove all the duplicate articles, and any that weren't removed automatically were manually removed.

### Eligibility criteria

We only included studies that fulfilled the following eligibility criteria:

1. Study design: observational studies (cross-sectional, longitudinal, case-control, cohort).

2. Population of interest: apparently healthy children and adolescents (12–18) year.

3. Measure of ST as an exposure: Included studies that reported type of ST (TV, computer, videogames, internet, smartphone, tablet) quantified duration/frequency either self-reported or observed measure.

4. Measure of MetS as an outcome: Included studies that reported standard definition and/or criteria used to establish MetS diagnosis.

5. Measure of relationship: examined association between ST and MetS as odds ratio (ORs) or equivalent with their 95% confidence interval (CI).

### Exclusion criteria

We excluded reviews in which ST was not defined adequately or where time spent on various forms of screens was not differentiated from other forms of sedentary lifestyle. Studies examining sedentary behaviour but reporting findings for ST separately from other forms of sedentary behaviours were included. Studies were excluded if MetS diagnosis was not defined adequately, not an observational study design, no reporting of ORs or equivalent, studies including adolescents with pathological conditions, population younger than 12 year or older than 18 years, and studies assessing relationship of ST with outcomes other than MetS such as obesity, physical inactivity, or cardiovascular risk.

## Study selection

Through systematic search, titles and abstracts were screened independently by two investigators (S.M and R.E), and potentially eligible articles were identified after removal of duplicates. Full text articles were retrieved for studies found to be relevant and compatible with eligibility criteria. Any discrepancies during the selection process were resolved either through consensus or consultation with the third investigator (I.D).

## Data extraction

Data extraction and full text review of eligible studies were cross-checked by two independent authors (S.M and R.E) for accuracy. A standardized data extraction table was created, including key characteristics of the identified studies as the following: descriptive study characteristics (author, publication year, country, study design, sample size, age, gender), screen type, exposure, and outcome indicator measures. Results were extracted as risk estimates: Odds Ratio or prevalence ratio with corresponding confidence intervals or z-score of MetS. A P-value of <0.05 was considered as a cut-off for statistical significance.

## Quality assessment

The National Institute of Health (NIH) Quality Assessment Tool for Observational Cohort and Cross-Sectional Studies was used to evaluate the risk of bias [35]. The checklist comprised 14 items for longitudinal research, of which only 11 could be applied to cross-sectional studies. Each item of methodological quality was classified as yes, no, or not reported and based on number of yes as total score, studies were classified according to quality rating: Poor<50%, Fair 50–75% and good >75%. Possible disagreements on the final score were resolved by consensus among the authors. Studies met from 73% to 91% of the quality criteria, with 9 studies (9/10, 90%) meeting good scoring indicating low risk of bias. All studies clearly stated the main aim, population and definition of exposure/outcome. However, two studies (2/10, 20%) did not use key potential confounders in the analysis. Eleven items were applicable to nine studies due to cross-sectional nature of these studies and one perspective cohort study where all 14 items were applicable (Table 1). Details of the NHL Quality assessment questions (Q1-14) are shown in S2 Table.

## Data analysis

Synthesis began by summarizing review results and conclusions in note form. Reviews were then grouped by the exposure, which is screen time, and the outcome of interest was measured, which is the MetS and related risk factors. Moreover, we examined the conclusions of the included studies to decide which article came out as plausible. However, we did not enumerate the findings across studies as quantitative summaries should be undertaken at an individual study level rather than at a review level. A descriptive analysis of each included publication was conducted. ST exposure in hrs/day or week and the observed prevalence of MetS in percentages were specified. Adjusted estimates of OR or MetS z-score for the association between ST and MetS with a corresponding 95% CI were obtained. The OR of the dose-response gradient effect was also extracted.

# Result

## Flow of the studies

Fig 1 displays a flow chart of study identification and selection. Systematic search on database identified 3521 abstracts; of these, 2137 were excluded during initial screening for

**Table 1. Study quality assessed by the quality assessment tool for observational cohort and cross-sectional studies.**

| Author | Items of Quality Assessment Tool for Observational Cohort and Cross-Sectional Studies | | | | | | | | | | | | | | |
|---|---|---|---|---|---|---|---|---|---|---|---|---|---|---|---|
| | 1 | 2 | 3 | 4 | 5 | 6 | 7 | 8 | 9 | 10 | 11 | 12 | 13 | 14 | Total score |
| Schaan et al. [36] | Y | Y | Y | Y | Y | Y | NA | Y | Y | N | Y | NA | NA | Y | 10/11 (91%) |
| Khan M et al. [32] | Y | Y | Y | Y | Y | N | NA | Y | Y | N | Y | NA | NA | Y | 9/11 (82%) |
| Mark E and Janssen [37] | Y | Y | Y | Y | Y | Y | NA | Y | Y | N | Y | NA | NA | Y | 10/11 (91%) |
| Kang HT et al. [38] | Y | Y | Y | Y | Y | Y | NA | Y | Y | N | Y | NA | NA | Y | 10/11 (91%) |
| de Oliveira RG et al. [31] | Y | Y | Y | Y | Y | Y | NA | Y | Y | N | Y | NA | NA | Y | 10/11 (91%) |
| Siwarom S et al. [39] | Y | Y | Y | Y | Y | N | NA | Y | Y | N | Y | NA | NA | Y | 9/11 (82%) |
| Hardy L et al. [40] | Y | Y | Y | Y | N | N | NA | Y | Y | N | Y | NA | NA | Y | 8/11 (73%) |
| Fadzlina A et al. [41] | Y | Y | Y | Y | Y | Y | NA | Y | Y | N | Y | NA | NA | N | 9/11 (82%) |
| Grøntved A et al. [42] | Y | Y | Y | Y | Y | Y | Y | Y | Y | Y | Y | N | N | Y | 12/14 (86%) |
| de Castro Silveira et al. [43] | Y | Y | Y | Y | Y | Y | NA | Y | Y | N | Y | NA | NA | Y | 10/11 (91%) |

**Total score**, number of yes; **NA** not applicable, **N**, not present, **Y**, present.

**Quality rating**: poor <50%, Fair 50–75%, Good >75%

unrelated topics, meeting the exclusion criteria and duplicate studies from different databases. Totally, 62 full text articles were assessed to examine their eligibility for inclusion in the current review, and finally, after review of the full texts, ten studies were included in the data extraction.

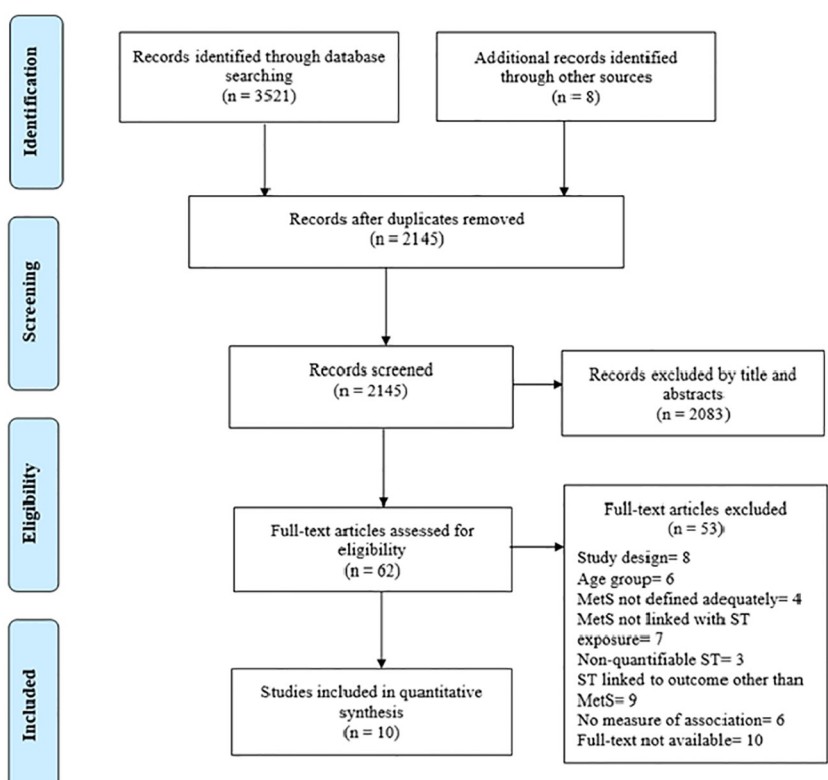

**Fig 1. PRISMA flowchart for review.**

## Overview of studies

Table 2 presents the general characteristics of the eligible studies. Most studies (9/10, 90%) employed a cross-sectional study design, except for one that implemented a prospective cohort design. In total, data from 41,687 participants was included in this review. The sample size ranged from 474 to 33,900 and the mean age of participants ranged from 12–18 years. Six of the studies have utilized school-based setting [31, 32, 36, 40–42], while the remaining 4 were part of national surveys. The primary objective of included studies was to establish significant association between ST (of any type) and MetS among adolescents, with all studies considering single or multiple exposures like PA, dietary habit or sleep duration associated with ST. Although the scope of the review focused on adolescents at the age of (12–18) year, studies that had a range of below 12 or above 18 were not excluded if the mean age was between 12–18 years. All the studies used subjective measures of daily/weekly hours ST including parental report, self-reported interview questionnaires. The main modalities of ST were TV viewing, computers, videogames, internet, tablets, and smartphones. Nine of the studies (9/10, 90%) utilized multi-variable analysis to adjust for covariates [31, 32, 36–40, 42, 43], 8 of which (8/10, 80%) [31, 32, 36–38, 40, 42, 43], found a positive association between ST and MetS. Seven studies (7/10, 70%) proved a dose-response gradient for that association [29, 30, 34–36, 40, 41]. Two studies by Fadzlina et al. [41] and Siwarom et al. [39] found no significant association between ST and MetS. Exposure to ST prior to the age of 2 years was significantly associated with MetS in one of the included studies [39].

## Screen time

All the studies have examined the measure of time in relation to different screen types, including TV viewing, computers, videogames, internet, tablets and smartphones. Daily or weekly reported hours of ST exposure was the main exposure measure in all studies. In most of the studies, the exposure variable was grouped into categories; two ($< 2$; $> 2$ hr/day), three ($\leq 2$, 3–5 and $\geq 6$h/day) or four subgroups ($\leq 1$, 2,3,4, $\geq 5$hr/day), ($<2$, 2, 4, $\leq 6$), (0–16, 17–24, 25–34, $\geq 35$ hr/week). Odds of MetS were compared according to each response category with least subgroup being as the reference. Seven of the studies have indicated a graded dose-response relationship between ST and MetS. Khan et al. [32] found that for every increased hour of ST, the risk of MetS is increased by 1.21. (1.08–1.35). Similarly, the cohort study of 12 years follow-up [42] showed that for each 1-hour increment in TV viewing time during adolescence, the MetS z-score is increased by 0.45 (95% CI 0.14 to 0.76). Screen media exposure during the first two years of life was independently linked to a 30% increase in MetS in adolescence (OR 1.3, 95% CI: 1.01–1.68) [42].

## Prevalence of MetS and ST dose-response effect

The prevalence of MetS among adolescents ranged from 2.6 percent to 17 percent in eight of the included studies (8/10, 80 percent), with a gradually rising trend toward longer ST duration. Three studies reported outcomes of interest as metabolic Z-score [42], multiple metabolic risk factors [40] or continuous metabolic risk score [43]. For instance, Hardy et al. [40] reported more than twofold increase in the risk of abnormal insulin levels (adjusted OR, 2.42; 95% CI, 1.11–5.28) or elevated HOMA-IR (adjusted OR, 2.42; 95% CI, 1.11–5.28) among boys exceeding 2 hours of ST per day. Grøntved et al. [42], on the other hand, utilized the MetS z-score, which was found to be 0.35 (95 percent CI: 0.08–0.62) and that each hour of TV viewing was associated with 0.45 increase in MetS z-score. de Castro Silveira et al. [43] found that exposure to ST $\geq$ 2h/day when adjusted for cardiorespiratory fitness yielded a significant association with metabolic risk score (High ST/unfit (1.07:1.01–1.13, p = 0.020), Low ST/unfit 1.08:

**Table 2. Summary of characteristics of included studies showing relation between screen time (ST) and Metabolic syndrome (MetS).**

| Author; publication year | Country | Study design; Sample size (N) | Mean age at baseline (SD); gender | Screen type | Exposure ST measure | Outcome measures (MetS) | Association with MetS OR (95%CI) | Comments |
|---|---|---|---|---|---|---|---|---|
| Schaan et al. (2019) [36] | Brazil | 33,900; cross-sectional | 14.6 year (SD not reported); 59.4% Female | TV view, computers, videogames | Self-reported hours per day | IDF guidelines (WC, SBP, DBP, Fasting blood glucose, Triglycerides; HDL) | ST ≥6 h/day; 1.68 (1.03–2.74). | Prevalence of MetS 2.6% (95%CI: 2.3–3.0), ST remained significantly associated with MetS after adjusting of covariates; age, sex, socioeconomic, PA. Association remained significant MetS remained significant only for adolescents who reported consumption of snacks in front of screens. |
| Khan M et al. (2019) [32] | UAE | 474; cross-sectional | 14.9 ±1.9 years; 47% Female | Computer, television, and video game | Self-reported hours per day | IDF guidelines (WC, SBP, DBP, Fasting blood glucose, Triglycerides; HDL) | ST ≥2 h/day: 2.20 (1.04–4.67) Each hour of increased ST (1.21; 1.08–1.35) | Prevalence of MetS 8.5% in <2hr/d, 13.4% ≥2 hr/d) Association was adjusted for age, sex, physical education classes, smoking, parental education, daily intake of carbonated drink, fruits, vegetables, milk, fast food |
| Mark E and Janssen (2008) [37] | US | 1803; cross-sectional | 15.9 ± 2.2 years; 50.3% Female | TV, video, computer game | Self-reported hours per day home interview/ mobile exam centre | NCEP ATP II: ≥3 of the following: high triglycerides, high fasting glucose, high WC, high BP, low HDL. | ST ≥5 h/day: 2.90 (1.39–6.02) | Prevalence of MetS 3.7% in ≤1 hr/d, 8.4% in ≥5 hr/day. Association was adjusted for age, smoking and PA. |
| Kang HT et al. (2010) [38] | Korea | 845, cross-sectional | 13.4 ± 2.5 years; 46.9% Female | TV time, computer game, internet | Self-reported hours per week | NCEP ATP II: ≥3 of the following: high triglycerides, high fasting glucose, high WC, high BP, low HDL. | ST (≥35 h/week: 2.23 (1.02–4.86) | Prevalence of MetS 7.3%. Association was adjusted for age, sex, household income, residence area. |
| de Oliveira RG et al. (2014) [31] | Brazil | 1,035, cross-sectional | Mean not reported; 56.6% of (12-15y), 43.4% of (16-20y), 54.6% Female | TV, computer, video game, tablet, smartphone | Self-reported hours per day | IDF guidelines (WC, SBP, DBP, Fasting blood glucose, Triglycerides; HDL) | ST> 2 h/day: 1.32 (1.07–1.94) | Prevalence of MetS 4.5% (95% CI: 3.8–5.4). Association was adjusted for demographic, anthropometric nutritional indicators and, lifestyle determinants. |
| Siwarom S et al. (2021) [39] | Thailand | 1934, cross-sectional | 13.40 ± 1.94; 49.7% Female | television watching, computer, smart phone, tablet use | Self-reported hours per week/ screen media exposure during the first 2 years of life | IDF, Cook's, and de Ferranti's. | MetS by 1 out of 3 definitions: Exposure to screen media during the first 2 years of life: 1.30 (1.01–1.68). No association between total ST & MetS: 1.00 (0.99–1.00) | Prevalence of MetS 17%, Association of ST and MetS was adjusted for age, sex, foot intake, fruits and vegetables, PA. |

(*Continued*)

**Table 2.** (Continued)

| Author; publication year | Country | Study design; Sample size (N) | Mean age at baseline (SD); gender | Screen type | Exposure ST measure | Outcome measures (MetS) | Association with MetS OR (95%CI) | Comments |
|---|---|---|---|---|---|---|---|---|
| Hardy L et al. (2010) [40] | Australia | 496, cross-sectional | 15.4 ± 0.4 year; 42% Female | watching television/DVDs/videos and using a computer for recreation | Self-reported hours per day. Adolescent Sedentary Activity Questionnaire | Metabolic risk factors: Insulin level Glucose level HOMA-IR, HDL-C, LDL-C, Triglyceride, hs-CRP, ALT, GGT l, SBP, DBP | ST ≥2 h/day Boys: HOMA-IR (adjusted OR, 2.42 (1.11–5.28), insulin levels (adjusted OR, 2.73 (1.43–5.23) Girls: no association | Prevalence of abnormal biomarker e.g., Insulin in ≥2h/d is 22.7% boys vs, 22.9% girls; HOMA-IR 41.5% boys vs. 46.5% girls. Association was adjusted for BMI, SES (IRDS score), EDNP food score, Tanner score, and CRE (number of laps) |
| Fadzlina A et al. (2014) [41] | Malaysia | 1014, cross-sectional | 12.88 ± 0.33 years; 61.8% Female | Not reported | Self-reported hours per day | IDF guidelines (WC, SBP, DBP, Fasting blood glucose, Triglycerides; HDL) | No association between ST and MetS | Prevalence of MetS 2.6%in total, 10% among overweight. Obese No adjusted model was utilized |
| Grøntved A et al. (2020) [42] | EYHS, Danish cohort | 435, cohort | 15.6 ± 0.4 year; 54.5% Female | TV, computer use | Self-reported hours per day | MetS z-score based on AHA/NHLBI; WC, SBP, DBP, triglycerides, HDL (inverted), fasting glucose, fasting insulin | Total ST > 2 h/day a/w MetS z-score. 0.35 (0.08–0.62) Each 1-hour increment in TV viewing time; syndrome z-score 0.45 (0.14–0.76) | MetS Z-score for ≤1h (−0.2 ± 2.6), 1-3h (−0.1 ± 2.5) >3 h were (1.2 ± 3.5) Adjusted for age, gender, cohort, parental education level, current smoking status, (MVPA), intake of soft drinks, fruit- and vegetable intake, and family history of cardiovascular disease. |
| de Castro Silveira et al. (2020) [43] | Brazil | 1200, cross-sectional | Up to 17 years, no mean age reported; 56% Female | Not reported | Self-reported hours per day | Continuous metabolic score (CMetS) > 1 as metabolic risk factor Z-score of WC, SBP, glucose, triglyceride, total cholesterol, LDL, HDL | ST ≥2 h/day; Prevalence Ratio (PR) = 0.99 (0.95–1.03), insignificant association | Prevalence of metabolic risk 14.7%. ST was adjusted for cardiorespiratory fitness measured at time of recruitment yielded significant association. |

Abbreviations: IDF, International Diabetes Federation, NCEP ATP II, National Cholesterol Education Program Adult Treatment Panel; SD, Standard deviation; WC, Waist circumference; SBP, systolic blood pressure, DBP, diastolic blood pressure, PA, physical activity, BMI, Body mass index; SES, socioeconomic status; IRDS, Australian Bureau of Statistics Index of Relative; EDNP, energy-dense nutrient-poor; CRE, cardiorespiratory endurance. EYHS, European Youth Heart Study; AHA, American Heart Association (AHA); NHLBI, and the National Heart, Lung, and Blood Institute. HDL-C, high density lipoprotein cholesterol, LDL-C, low density lipoprotein cholesterol; HOMA-IR, Homeostatic Model Assessment for Insulin Resistance; h-s CRP, high sensitivity C-reactive protein; ALT, Alanine Aminotransferase; GGT, Gamma-Glutamyl Transferase.

1.02–1.14, p = 0.011) as compared to unadjusted ST (0.99: 0.95–1.03, P = 0.645). Metabolic risk was also higher in those with low ST/unfit (8%) and high ST/unfit (10%). (7 percent). Khan et al. [32] in their study found that higher prevalence of MetS was noted amongst adolescents who spent two hours or more of ST as compared to those with less than two hours (13.4 percent vs. 8.5 percent), respectively. The study also reported that for each hour increase in ST, the risk of MetS was increased by 21 percent (OR, 1.21; 95% CI: 1.08–1.35). Adolescents who spend more than two hours per day of ST were two times more likely to develop MetS (aOR = 2.20; 95% CI: 1.04–4.67) as compared to those who spend less than 2 hours/day. Concerning the dose-response gradient, six of the included studies have confirmed this hypothesis

with 5–6 hours of ST yielding the highest odds of MetS. While most studies utilized a 2-hour cut-off point, for greater accuracy, further subcategorization was performed during the multivariate analysis. In two comparable studies, a twofold increase in the odds of MetS was observed among adolescents who spent more than two hours [34] and five hours per day [40], respectively.

## Metabolic syndrome

Seven studies (7/10, 70%) [31, 32, 36–39, 41] used one of the following MetS outcome measure definitions: IDF, NCEP ATP II, Cook's, or de Ferranti's. Hardy et al. [40], on the other hand, classified metabolic risk factors including insulin level, glucose level, HOMA-IR, HDL-C, LDL-C, triglyceride, hs-CRP, ALT, GGT, SBP and DBP as isolated outcomes rather than MetS diagnosis. In contrast, in the cohort study of Grøntved et al. [42], the outcome was calculated as a continuous MetS z-score to preserve statistical power and because the number of incident cases of MetS according to the American Heart Association (AHA) and the National Heart, Lung, and Blood Institute (NHLBI) definitions with the additional inclusion of fasting insulin was calculated as a continuous MetS z-score. Furthermore, de Castro Silveira et al. [43] used metabolic risk assessment, which was calculated by adding the Z score of the following parameters: WC, SBP, glucose, triglycerides, total cholesterol, LDL and HDL, where MetS values greater than 1 were considered metabolic risk.

## Adjustment of covariates

Age, sex, socioeconomic level, region of residence, physical education classes, cardiorespiratory fitness, nutritional status, smoking, parental education, and BMI were all corrected for in nine of the studies (9/10, 90 percent) [31, 32, 36–40, 42, 43]. Even though seven of the studies [31, 32, 36, 37, 40, 42, 43] computed ORs adjusted for PA level and/or nutritional status, ST remained an independent determinant of MetS, apart from one study where unadjusted ST was found to be an insignificant predictor of metabolic risk [43]. In multivariate analysis, the relationship between screen-based sedentary behaviour and MetS remained significant only for teenagers who conveyed snacking in front of the screens [36].

## Discussion

Public health measures related to the COVID-19 pandemic have led to a critical increase in the use of digital screen devices and reliance on remote learning. Screen-based sedentary behaviour is linked to physical inactivity and increased caloric consumption, which are important contributors to obesity and cardio-metabolic risk. Taken together, a better understanding of the association between ST (of different types) and MetS among vulnerable populations i.e., adolescents, is necessary to target preventable causes of premature mortality in later adulthood.

The present systematic review provides a narrative synthesis of data concerning the relationship between ST and MetS among adolescents. The majority of studies indicate a positive and dose-response association between exposure and the outcome of interest. Based on Quality Assessment Tool for Observational and Cross-Sectional Studies, 90 percent of included studies indicated low risk of bias, demonstrating good quality score.

Findings from this review confirm that adolescents engaged in screen-based sedentary behaviour have an increased likelihood of developing MetS. Multivariate analysis demonstrated that ST was a significant independent predictor of MetS. These findings are consistent with Tremblay et al. [44], who reported in their systematic review and meta-analysis of 232 publications a positive correlation between increased levels of sedentary behaviour (especially

TV viewing> 2h/day) and cardiometabolic illnesses among children and youth. A notable link was observed between ST and adverse body composition, poor fitness, and low self-esteem [44]. Contrarily, some scholars have argued that ST in adolescents is irrelevant to future health risks. A systematic review [31] showed weak evidence of an association between ST and poor cardiorespiratory fitness, poorer cognitive development, lower educational attainments, poor sleep outcomes, or risk of MetS. However, it is important to note that weak association does not imply absence of correlation. For instance, the authors have reported a lack of literature as the probable cause of such an observation.

Time spent in front of screens (of different types), whether at home or school-based, was obtained as a self-reported measure based on daily/weekly hours. Longitudinal studies [42, 45] established that longer duration or more frequent TV viewing was associated with a higher clustering cardio-metabolic risk score, particularly elevated systolic blood pressure. In contrast, computer-based ST was associated with higher diastolic blood pressure, while a lower level of HDL was objectively associated with longer-accelerometer-derived sedentary time [46]. Sedentary behaviour was not found to be associated with other cardiometabolic risk factors such as triglycerides, HOMA-IR, or glucose level [42].

This systematic review indicates a linear association between ST and MetS, meaning that MetS risk increased in tandem. For instance, spending more than two hours of daily ST triggered an increased risk of MetS in a dose-response manner, with the most harmful effect noted at 5–6 hours per day. Consistent with our findings, longitudinal studies have shown that higher cholesterol levels [47] and higher blood pressure [48] were associated with watching more than two hours of television per day, as compared to those who watch less. Similarly, high levels of self-reported sedentary behaviour including ST were associated with an increased risk of elevated systolic and diastolic blood pressure [48–51], higher glycated haemoglobin (HbA1C) [52], fasting insulin [50, 53], insulin resistance [54], and MetS [55]. With reference to the dose-response gradient, longer duration of TV viewing was significantly correlated with increased risk of MetS/cardiovascular disease risk factors, whether a 2-h cut-point [56, 57], a 3-h [58] or a 4-h [59] cut-point [60] was utilized. Accordingly, adolescents should be urged to limit their daily recreational ST to less than 2 hours per day, as recommended by the American Academy of Paediatrics and the World Health Organization (WHO). This is especially crucial given the likelihood of such a risk persisting into adulthood.

Several studies have conveyed a rising trend of ST and digital technology use during the pandemic [33, 61]. Children and adolescents are considered amongst the most susceptible groups due to their limited self-regulation and liability to peer pressure. For instance, Xiang et al. [62], illustrated a considerable increase in ST during the pandemic among 6–17 years old in Shanghai (+1730 minutes or nearly 30 hours in total). More than a quarter of students revealed an increase in leisure-based ST in addition. Comparably, the median time spent in PA was shown to be reduced from, 540 minutes per week (prior to the pandemic) to 105 minutes per week, with an average reduction of 435 minutes. The prevalence of physical inactivity was almost tripled as compared to the pre-pandemic period (21.3 vs. 65.6%). Another study [63] conducted in Canada exhibited a significant decline of adherence into PA and ST guidelines among 5–17-year-old during the pandemic (PA; from 18% to 35% [64] and ST: from 64% to 11.2% [65], subsequently).

Despite the well-established harmful effects of ST, several academic studies have indicated possible benefits (especially educational outcomes) of ST among younger generations. According to several studies, children and teenagers frequently lack the discipline and insight to limit ST on their own [65]. Thus, taking into account the current challenging crisis [66], it becomes critically vital to address mitigation strategies and encourage a family-centred media use plan

that allows us to balance the risks and benefits. Despite the fact that ST has been traditionally linked to sedentary behaviour and increases the risk of negative health outcomes [67], there are always alternative positive ways around the corner. Appropriately used, ST can promote PA during a shelter-in-place [68], such as online PA classes, workout apps for mobile devices or active video games [68]. In this way, a recent systematic review among adolescents found that digital interventions incorporating educational activities, goal setting, self-monitoring, and parental involvement have led to a significant increase in PA [69].

Evidence is validating the role of opportunistic behaviour change counselling by clinical practitioners in the management of sedentary behaviour. A key mitigative strategy is to identify correlates of sedentary time, such as sociodemographic attributes, accessibility, parental behaviour, psychological, PA, and dietary behaviour [70]. Engaging adolescents, families, schools, and social workers in healthy lifestyle choices provides an enabling environment that supports behavioural change. School could be an ideal setting for encouraging physical activity and reducing sedentary behaviour. For instance, Verloigne et al. [71] acclaimed the placing of standing desks in classrooms and promoting stand-up bouts as an interventional approach to increase pupils' self-efficacy and reduce sedentary behaviour in a pleasurable manner.

It may be possible to use a holistic method that looks at all three levels of the socio-ecological model (intra-individual, inter-individual, physical environment, and policy) to help people move more and make healthy food choices [72]. Effective communication with adolescents and their families enhances digital literacy related to screen types, content, and setting screen limits. A systematic review and meta-analysis on the effect of behavioural interventions in reducing ST revealed that smaller sample sizes and shorter intervention durations were associated with greater impact. Involvement of healthcare professionals in setting goals, feedback, and planning clusters yielded better outcomes in ST reduction [73].

Healthcare workers are advised to initiate early prevention strategies tackling the associated risk factors of NCDs among adolescents. Given the challenging period of COVID-19 [74–78], it has become increasingly important to integrate lifestyle education, health promotion, and community awareness within the management. Screening for early identification of behavioural and metabolic risk factors will help reduce the burden of NCDs later in adulthood. Tools such as screening of baseline PA, ST, Body Mass Index (BMI) and psychological assessment are essential to identify modifiable risk factors and 'at risk' children. Accordingly, replacement strategies, weight loss programs, and exercise prescriptions are recommended, taking into account the importance of continuous monitoring and evaluation. Evidence from this study can guide national and public health efforts in planning accessible multisectoral prevention and intervention strategies to tackle the determinants of MetS and NCDs.

## Limitations

The evidence in this review was dependent on peer-reviewed journals via scientific databases, not accessing data from unpublished reports from educational institutions, non-profit data, or community services, and maybe subject to publication bias. Due to the apparent high risk of bias (ROB) attributable to clinical and methodological heterogeneity, meta-analysis of data was not performed. Moreover, undefined information in any of the included studies was not confirmed by the related authors, jeopardizing their quality, if any important details were missed. Furthermore, our research was limited to studies published only in English, and the cross-sectional design of the majority of included studies prevented inference of causality, thereby limiting the conclusion drawn regarding the temporal relationship between ST and MetS. Data were gathered by one researcher, and even though the data were carefully checked back to the publication by the second researcher, we did not use two separate extractions. In

our narrative synthesis of findings, we aimed to avoid vote-counting of numbers of positive or negative studies to judge the strength of evidence. However, it is possible that our findings reflect methodological or conceptual biases in our included reviews. Finally, the search did not extend to all existing databases. Nonetheless, we performed searches in two primary databases and one secondary database.

## Conclusion

The COVID-19 pandemic has resulted in children and adolescents spending more time on digital screen devices, eliciting a profound effect on their cardio-metabolic health and NCDs burden. In brief, our review demonstrated that independent of PA, significant association between ST and MetS was noted among adolescents. This observation has significant public health and clinical implications that demand urgent prevention initiatives targeting young people and their parents. Such interventions aim to enhance early screening of behavioural and metabolic risk factors and increase awareness of potential adverse health impacts related to NCDs. Healthcare providers should consider a promotive, holistic approach, taking into consideration the international recommendation of ST and PA across different age groups. Further community-based research, including longitudinal and RCTs are needed to confirm this primarily observational evidence.

## Supporting information

**S1 Table. PRISMA 2020 checklist.**
(PDF)

**S2 Table. NIH quality assessment tool for observational and cross-sectional studies.**
(PDF)

## Author Contributions

**Conceptualization:** Sarah Musa, Rowaida Elyamani, Ismail Dergaa.

**Data curation:** Sarah Musa, Ismail Dergaa.

**Formal analysis:** Sarah Musa.

**Investigation:** Sarah Musa, Rowaida Elyamani.

**Methodology:** Sarah Musa, Rowaida Elyamani.

**Project administration:** Ismail Dergaa.

**Supervision:** Ismail Dergaa.

**Validation:** Sarah Musa, Rowaida Elyamani.

**Visualization:** Sarah Musa.

**Writing – original draft:** Sarah Musa, Rowaida Elyamani.

**Writing – review & editing:** Sarah Musa, Rowaida Elyamani, Ismail Dergaa.

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
