## [Decision Letter · Decision Letter 0]

2 Feb 2022

PONE-D-21-30881COVID-19 and screen-based sedentary behaviour: Systematic review of digital screen time and metabolic syndrome in adolescentsPLOS ONE

Dear Dr. Sarah,

Thank you for submitting your manuscript to PLOS ONE. After careful consideration, we feel that it has merit but does not fully meet PLOS ONE’s publication criteria as it currently stands. Therefore, we invite you to submit a revised version of the manuscript that addresses the points raised during the review process.

ACADEMIC EDITOR:Dear author/s,Thank you for submitting your work with us. Your manuscript cannot be accepted in its present form. Please attends to the reviewer comments and improve on them.  The decision of this manuscript is justified on PLOS ONE’s publication criteria and not on its novelty or perceived impact.

We look forward to receiving your revised manuscript.

Kind regards,

Zulkarnain Jaafar

Academic Editor

PLOS ONE

Journal Requirements:

2. PLOS ONE does not copy edit accepted manuscripts (https://journals.plos.org/plosone/s/criteria-for-publication#loc-5). To that effect, please ensure that your submission is free of typos and grammatical errors.

*Please ensure that your PRISMA flowchart details the reasons for record exclusion at each step.

3. Thank you for stating the following in the Acknowledgments/ Funding Section of your manuscript: 

Open access is funded by Qatar National Library (QNL)

7. We noticed you have some minor occurrence of overlapping text with the following previous publication(s), which needs to be addressed:

- https://pubmed.ncbi.nlm.nih.gov/27997601/

- https://bmjopen.bmj.com/content/bmjopen/9/1/e023191.full.pdf?ck_subscriber_id=31674

- https://ijbnpa.biomedcentral.com/articles/10.1186/1479-5868-8-98

- https://link.springer.com/article/10.1186/s12966-018-0726-9

In your revision ensure you cite all your sources (including your own works), and quote or rephrase any duplicated text outside the methods section. Further consideration is dependent on these concerns being addressed.

Reviewers' comments:

Reviewer's Responses to Questions

**Comments to the Author**

1. Is the manuscript technically sound, and do the data support the conclusions?

Reviewer #1: Partly

Reviewer #2: Yes

2. Has the statistical analysis been performed appropriately and rigorously? 

Reviewer #1: N/A

Reviewer #2: Yes

3. Have the authors made all data underlying the findings in their manuscript fully available?

Reviewer #1: No

Reviewer #2: Yes

4. Is the manuscript presented in an intelligible fashion and written in standard English?

Reviewer #1: Yes

Reviewer #2: Yes

5. Review Comments to the Author

Reviewer #1: There are major revisions needed in the manuscript.

-The introduction should be written in more detail and clearly.

-The method should be better communicated to readers.Statistical analyzes should be expressed more clearly.

-Expand the discussion section and provide advice for practical practitioners.

Reviewer #2: This review is topical and will add to the evidence to support the need for interventions to reduce sedentary behaviour across the age spectrum. The authors have presented recent data to improve research in an important area.

6. PLOS authors have the option to publish the peer review history of their article (what does this mean?). If published, this will include your full peer review and any attached files.

Reviewer #1: **Yes: **Zeki AKYILDIZ

Reviewer #2: No

---

## [Author Response · Author response to Decision Letter 0]

1 Mar 2022

TO REVIEWER #1

Dear Reviewer,

Thank you for your comments. Please find below the responses to your questions/suggestions. Sincerely yours.

1. There are major revisions needed in the manuscript.

Thank you for your remark. All your suggestions have been taken into considerations and changes made to full text accordingly.

2. The introduction should be written in more detail and clearly.

Thank you for your remark. We ado agree that a brief introduction section is advantageous to provide more context. Changes have been done to the introduction part of the main text according to your suggestion. The following sentences were added:

Modification: (L80-84)

“Given the revolutionary advances in digital technologies, the question of how to adequately classify ST remains a challenge [10]. The World Health Organization (WHO) defines ST as “Time spent passively watching screen-based entertainment (TV, computers, mobile devices)," excluding other innovative and modern forms of virtual realities, interactive video-gaming where physical activity or movement is required [11]”

Modification: (L85-93)

“The COVID-19 pandemic has caused a marked increase in ST across the globe. A large observational study (n=8395) in 10 European countries revealed that 69.5% [95%CI: 68.5- 70.5] of young adolescents aged 6-18 years have exceeded the recommended limit of ST (>2 h/day) during weekdays and 63.8% during weekend [95%CI: 62.7- 64.8]. Children residing in mildly affected countries and those in countries with lower level of restrictions were less likely to exceed that limit (OR= 3.25 [95%CI: 2.38 - 4.45) and OR= 1.42 [95% CI: 1.07-1.90], respectively) [12]. Similarly, findings from (ABCD) study during the early stages of the pandemic reported a mean (SD) of 7.70 (5.74) h/day of screen use, a more than twofold increase as compared to the pre-pandemic figure [13].”

Modification: (L98-109)

“According to the American Heart Association (AHA), National Heart, Lung and Blood Institute (NHLBI) and International Diabetes Federation (IDF), the diagnosis of MetS is based on the presence of three of the followings: waist circumference (WC) indicative of central obesity (at least 102 cm in men and 89 cm in women), raised triglyceride (<40 mg/dl) in males, <50 mg/dl in females), raised blood pressure (systolic BP≥ 130 or diastolic BP ≥ 85 mmHg or receiving treatment for hypertension), and raised fasting glucose level (≥100 mg/dL, or diagnosed with type 2 diabetes) [16]. The diagnosis of MetS is usually established after the age of 10 years. In older children and adolescents aged 10-16 years, MetS is diagnosed in the presence of central adiposity (≥90th) and two of the following: triglycerides (TG)≥ 150 mg/dl, HDL-C <40 mg/dl, systolic blood pressure (SBP) ≥ 130 mmHg or diastolic blood pressure (DBP) ≥85 mmHg, fasting plasma glucose (FG) ≥ 100 mg/dl or previously diagnosed type 2 diabetes [17]. “

Modification: (L110-112)

“MetS in children and adolescents has become a major public health concern with prevalence reaching as high as 38.9% in the general population and relatively higher in overweight/obese children [18].”

Modification: (L114-120)

“The pathological process underlying MetS begins already in childhood with complex interrelated genetic and environmental factors [22]. Evidence suggests that screen-based behaviours and physical inactivity are associated with higher levels of inflammatory biomarkers such as interleukin-6 (IL-6) and tumour necrosis factor-α (TNF-α) which stimulate C-reactive protein (CRP), an important causative pathway leading to dyslipidaemia, insulin resistance and cardiovascular diseases [23]. A study by Strizich et al., showed that lower levels of MVPA were associated with higher glucose/lipid profile, and increased inflammatory biomarkers [24].”

Modification: (L126-131)

“The association between ST and MetS among adolescents has been investigated in several studies prior to the declaration of the COVID-19 pandemic [26-27]. However, results were found inconclusive for the most part owing to limited data and generalizability of findings to different types of ST considering the duration, content, and context of exposure [28-29]. In a recent systematic review, authors pointed out limitations in approving the direct cause and effect relationship between excessive ST and MetS in adolescents [30].”

3. The method should be better communicated to readers. Statistical analyzes should be expressed more clearly. 

Thank you for highlighting this point. We've added a section entitled (Data analysis) to the main text. The following section was added:

Modification: (L223-232)

Data analysis

“Synthesis began by summarizing review results and conclusions in note form. Reviews were then grouped by the exposure, which is screen time, and the outcome of interest was measured, which is the MetS and related risk factors. Moreover, we examined the conclusions of the included studies to decide which article came out as plausible. However, we did not enumerate the findings across studies as quantitative summaries should be undertaken at an individual study level rather than at a review level. A descriptive analysis of each included publication was conducted. ST exposure in hrs/day or week and the observed prevalence of MetS in percentages were specified. Adjusted estimates of OR or MetS z-score for the association between ST and MetS with a corresponding 95% CI were obtained. The OR of the dose-response gradient effect was also extracted.”

3. Expand the discussion section and provide advice for practical practitioners. 

Thank you for the constructive recommendations.

 As acknowledged by the reviewer, changes have been done to the main text according to your suggestion. Implications of our review on home, school and clinical practice were added. The following details were added:

Modification: (L349-354) 

“Contrarily, some scholars have argued that ST in adolescents is irrelevant to future health risks. A systematic review [31] showed weak evidence of an association between ST and poor cardiorespiratory fitness, poorer cognitive development, lower educational attainments, poor sleep outcomes, or risk of MeTs. However, it is important to note that weak association does not imply absence of correlation. For instance, the authors have reported a lack of literature as the probable cause of such an observation.”

Modification: (L401-428)

“Evidence is validating the role of opportunistic behaviour change counselling by clinical practitioners in the management of sedentary behaviour. A key mitigative strategy is to identify correlates of sedentary time, such as sociodemographic attributes, accessibility, parental behaviour, psychological, PA, and dietary behaviour [70]. Engaging adolescents, families, schools, and social workers in healthy lifestyle choices provides an enabling environment that supports behavioural change. School could be an ideal setting for encouraging physical activity and reducing sedentary behaviour. For instance, Verlogine et al. [71] acclaimed the placing of standing desks in classrooms and promoting stand-up bouts as an interventional approach to increase pupils’ self-efficacy and reduce sedentary behaviour in a pleasurable manner.

It may be possible to use a holistic method that looks at all three levels of the socio-ecological model (intra-individual, inter-individual, physical environment, and policy) to help people move more and make healthy food choices [72]. Effective communication with adolescents and their families enhances digital literacy related to screen types, content, and setting screen limits. A systematic review and meta-analysis on the effect of behavioural interventions in reducing ST revealed that smaller sample sizes and shorter intervention durations were associated with greater impact. Involvement of healthcare professionals in setting goals, feedback, and planning clusters yielded better outcomes in ST reduction [73].

Healthcare workers are advised to initiate early prevention strategies tackling the associated risk factors of NCDs among adolescents. Given the challenging period of COVID-19, it has become increasingly important to integrate lifestyle education, health promotion, and community awareness within the management. Screening for early identification of behavioural and metabolic risk factors will help reduce the burden of NCDs later in adulthood. Tools such as screening of baseline PA, ST, Body Mass Index (BMI) and psychological assessment are essential to identify modifiable risk factors and ‘at risk’ children. Accordingly, replacement strategies, weight loss programs, and exercise prescriptions are recommended, taking into account the importance of continuous monitoring and evaluation. Evidence from this study can guide national and public health efforts in planning accessible multisectoral prevention and intervention strategies to tackle the determinants of MetS and NCDs.”

TO REVIEWER #2

Dear Reviewer,

Thank you for your comments. Please find below the responses to your questions/suggestions. Sincerely yours.

1.This review is topical and will add to the evidence to support the need for interventions to reduce sedentary behaviour across the age spectrum. The authors have presented recent data to improve research in an important area. 

Thank you for your positive feedback. We do appreciate your time and efforts reviewing our manuscript.

---

## [Decision Letter · Decision Letter 1]

4 Mar 2022

COVID-19 and screen-based sedentary behaviour: Systematic review of digital screen time and metabolic syndrome in adolescents

PONE-D-21-30881R1

Dear Dr.Musa,

We’re pleased to inform you that your manuscript has been judged scientifically suitable for publication and will be formally accepted for publication once it meets all outstanding technical requirements.

Kind regards,

Zulkarnain Jaafar

Academic Editor

PLOS ONE

Additional Editor Comments (optional):

Dear Authors,

Thank you for sharing your work with us. Congratulations that your manuscript is being accepted for publication. Hope to see you work again in the future.

Reviewers' comments:

Reviewer's Responses to Questions

**Comments to the Author**

1. If the authors have adequately addressed your comments raised in a previous round of review and you feel that this manuscript is now acceptable for publication, you may indicate that here to bypass the “Comments to the Author” section, enter your conflict of interest statement in the “Confidential to Editor” section, and submit your "Accept" recommendation.

Reviewer #1: All comments have been addressed

2. Is the manuscript technically sound, and do the data support the conclusions?

Reviewer #1: Yes

3. Has the statistical analysis been performed appropriately and rigorously? 

Reviewer #1: Yes

4. Have the authors made all data underlying the findings in their manuscript fully available?

Reviewer #1: No

5. Is the manuscript presented in an intelligible fashion and written in standard English?

Reviewer #1: Yes

6. Review Comments to the Author

Reviewer #1: Researchers have improved the article. It is acceptable as it is.

Congratulations to the authors for their contribution to the research.

7. PLOS authors have the option to publish the peer review history of their article (what does this mean?). If published, this will include your full peer review and any attached files.

Reviewer #1: **Yes: **Zeki Akyildiz

---

## [Editor Report · Acceptance letter]

11 Mar 2022

PONE-D-21-30881R1 

COVID-19 and screen-based sedentary behaviour: Systematic review of digital screen time and metabolic syndrome in adolescents 

Dear Dr. Musa:

I'm pleased to inform you that your manuscript has been deemed suitable for publication in PLOS ONE. Congratulations! Your manuscript is now with our production department. 

Kind regards, 

on behalf of

Dr. Zulkarnain Jaafar 

Academic Editor

PLOS ONE